# Antibiotic Susceptibility of *Staphylococcus aureus* and *Streptococcus pneumoniae* Isolates from the Nasopharynx of Febrile Children under 5 Years in Nanoro, Burkina Faso

**DOI:** 10.3390/antibiotics10040444

**Published:** 2021-04-15

**Authors:** Massa dit Achille Bonko, Palpouguini Lompo, Marc Christian Tahita, Francois Kiemde, Ibrahima Karama, Athanase M. Somé, Petra F. Mens, Sandra Menting, Halidou Tinto, Henk D. F. H. Schallig

**Affiliations:** 1Institut de Recherche en Science de la Santé, Direction Régionale du Centre-Ouest/Unité de Recherche Clinique de Nanoro, Nanoro 18, Burkina Faso; palpouguini.lompo@crun.bf (P.L.); marctahita@yahoo.fr (M.C.T.); francois.kiemde@crun.bf (F.K.); karama.ibrahima@crun.bf (I.K.); athanasesome@yahoo.fr (A.M.S.); tintoh@crun.bf (H.T.); 2Amsterdam University Medical Centers, Department of Medical Microbiology, Experimental Parasitology Unit, Academic Medical Center at the University of Amsterdam, 1105 AZ Amsterdam, The Netherlands; p.f.mens@amsterdamumc.nl (P.F.M.); s.menting@amsterdamumc.nl (S.M.); h.d.schallig@amsterdamumc.nl (H.D.F.H.S.)

**Keywords:** antibiotics, resistance, nasopharynx, children, *Streptococcus pneumoniae*, *Staphylococcus aureus*

## Abstract

(1) Background: nasopharynx colonization by resistant *Staphylococcus aureus* and *Streptococcus pneumoniae* can lead to serious diseases. Emerging resistance to antibiotics commonly used to treat infections due to these pathogens poses a serious threat to the health system. The present study aimed to determine the antibiotic susceptibility of *S. aureus* and *S. pneumoniae* isolates from the febrile children’s nasopharynx under 5 years in Nanoro (Burkina Faso). (2) Methods: bacterial isolates were identified from nasopharyngeal swabs prospectively collected from 629 febrile children. Antibiotic susceptibility of *S. aureus* and *S. pneumoniae* isolates was assessed by Kirby–Bauer method and results were interpreted according to the Clinical and Laboratory Standard Institute guidelines. (3) Results: bacterial colonization was confirmed in 154 (24.5%) of children of whom 96.1% carried *S. aureus*, 3.2% had *S. pneumoniae*, and 0.6% carried both bacteria. *S. aureus* isolates showed alarming resistance to penicillin (96.0%) and *S. pneumoniae* was highly resistant to tetracycline (100%) and trimethoprim–sulfamethoxazole (83.3%), and moderately resistant to penicillin (50.0%). Furthermore, 4.0% of S. aureus identified were methicillin resistant. (4) Conclusion: this study showed concerning resistance rates to antibiotics to treat suspected bacterial respiratory tract infections. The work highlights the necessity to implement continuous antibiotic resistance surveillance.

## 1. Introduction

Bacterial colonization of the nasopharynx in human can lead to the development of invasive and non-invasive disease, caused by common pathogens, such as *Staphylococcus aureus* and *Streptococcus pneumoniae*. Although nasopharyngeal carriage with *S. pneumoniae* and *S. aureus* is usually asymptomatic, it can lead to serious infections in children, such as pneumonia, sepsis, and otitis [1,2,3]. Moreover, the nasopharynx of healthy individuals is a potential reservoir for transmission of *S. pneumoniae* and *S. aureus* to other people in the community or health care setting [2,3,4,5]. Therefore, emerging resistance to commonly used antibiotics to treat infections caused by these bacteria is a serious threat to health systems [5,6,7,8,9]. This situation can lead to treatment failures, extended hospitalization, increased health care costs, and may ultimately lead to increased mortality and morbidity [8,9].

In Burkina Faso, *S. pneumoniae* became the leading cause of bacterial meningitis after the introduction of the *Haemophilus influenzae* type b vaccine in 2006 [10] and the serogroup A meningococcal conjugate vaccine (MenAfriVac) in 2010 [11]. However, the introduction of the thirteen-valent pneumococcal conjugate vaccine (PCV13) resulted in a significant decrease of invasive diseases caused by related strains, such as serotypes 6A/6B, 5, 14, 23F, and 18C/18F/B/18A, in children under 5 years of age in the country [12,13]. A reduction of around 50% in absolute number of cases of confirmed pneumococcal meningitis in children under 5 years was observed from the pre-PCV13 period (2011–2013; 478 confirmed pneumococcal meningitis cases) to the post-PCV13 period (2014–2015; 212 confirmed pneumococcal meningitis cases) [12,13]. Similarly, it was reported that the introduction of the pneumococcal conjugate vaccine into routine infant immunization programs substantially decreased invasive pneumococcal diseases in some other African countries [14,15]. 

Despite this reduction of pneumococci infections, some pneumococci genotypes resistant to antibiotics have emerged worldwide mainly in commensal micro flora [7,16]. In addition, it has been reported that pneumococcal conjugate vaccination might alter the upper respiratory tract flora and subsequently increase the risk of *S. aureus* colonization and diseases, particularly with methicillin-resistant *S. aureus* (MRSA) [17]. Although not extensively studied in Burkina Faso, reports from the West Africa region highlight a significant spreading of MRSA strains [18,19,20,21]. 

Improved insight in emerging bacterial resistance could be obtained when more antimicrobial resistance prevalence studies on nasopharyngeal carriage are conducted in febrile children under 5 years of age. This would improve monitoring and control of these emerging resistant bacteria and would also help to save lives of many children under 5. Bacterial colonization of the nasopharynx could be a proxy to assess bacterial resistance and pneumococcal serotype distribution [22]. Furthermore, the inter-human and environmental transmission of resistant strains are important determinants in the spread of bacteria resistant to antibiotics [23]. Consequently, studying potential pathogens of the nasopharynx can be a substantial add-on value to the antibiotic stewardship and antimicrobial resistance surveillance. Therefore, the present study aimed to determine the antibiotic susceptibility profile of *S. aureus* and *S. pneumoniae* isolates from the nasopharynx of febrile children under 5 years in Nanoro, Burkina Faso.

## 2. Results

### 2.1. Characteristics of Study Population

The characteristics of the study population are presented in Table 1 In total, 629 nasopharyngeal swabs were obtained from febrile children under 5 years. A significantly higher proportion (5% significance level; 1 degree of freedom) of males (54.1%; 340/629) were recruited. The median age of the enrolled children was 19 months (Interquartile range (IQR): 11.0–32.0). A significantly higher proportion (5% significance level; 1 degree of freedom) of the enrolled children (71.9%; 452/629) were infants (between 1 and 30 months of age) and a portion of this age group (30.1%; 136/452) did not receive pneumococcal vaccination according to the expanded national vaccination (EPI) program of Burkina Faso. 

Bacterial colonization of the nasopharynx was confirmed in 154 (24.5%) of the 629 febrile children. In total, 155 bacterial isolates were identified: 96.1% (148/154) children carried *S. aureus*, 3.2% (5/154) children had *S. pneumoniae*, and only one child, 0.6% (1/154) carried both bacteria (Table 1). The majority of isolates, 64.5% (100/155) were identified in the vaccinated children group, of which, 51.6% (83/155) were infants and 11.0% (17/155) were older toddlers (Table 1). Bacterial colonization was not observed in the single neonate recruited for the study and further data of this child are not presented in this paper.

Furthermore, bacterial colonization was not significantly different between gender (*p* = 0.55) and age groups of vaccinated children (*p* = 0.08) 

### 2.2. Antibiotic Susceptibility Testing (AST)

The result of AST of *S. aureus* isolates (*n* = 149) and *S. pneumoniae* isolates (*n* = 6) is reported in Table 2. Isolates from *S. aureus* and *S. pneumoniae* showed resistance rates of more than 85.0% to tetracycline (TET) (Table 2). Furthermore, *S. aureus* isolates showed a high resistance rate to penicillin (PEN) (96.0%; 143/149), but low resistance rates to trimethoprim-sulfamethoxazole (SXT) (14.8%; 22/149), to erythromycin (ERY) (14.1%; 21/149), to clindamycin (CC) (10.1%; 15/149), and to ciprofloxacin (CIP) (4.7%; 7/149). A very low resistance rate was observed against chloramphenicol (CL) (1.3%; 2/149). Furthermore, six (6) out of 149 *S. aureus* (4.0%) isolated were methicillin resistant (OX resistant), of which, five (5) (83.3%) were resistant to TET and PEN. All *S. aureus* isolates were susceptible to gentamicin (GEN) and vancomycin (VAN). 

The distribution of antibiotic susceptibility patterns of the 149 *S. aureus* isolates according to the different age groups is presented in Table 2. No difference was reported in the resistance of *S. aureus* to PEN and TET in both age groups. However, a notable higher resistance rate (24.3%; 9/37) for SXT was found in older toddlers compared to infants (11.6%; 13/112). In contrast, *S. aureus* identified in infants were more resistant to ERY (17.0%) and to CC (12.5%) than isolates recovered in older toddlers (5.4% or 2.7%, respectively). In both age groups, low resistance rates to CIP were observed, 4.5% in infants and 5.4% in older toddlers. In addition, all MRSA were identified in children older than one month. 

Next to 100% resistance to TET, all *S. pneumoniae* isolates (*n* = 6) showed high resistance to SXT 83.3% (5/6), medium resistance to PEN 50% (3/6), and only one isolate was resistant to chloramphenicol (CL) 16.7% (1/6) (Table 2). All *S. pneumoniae* isolates were 100% susceptible to ampicillin (AMP), and ceftriaxone (CRO). 

The distribution of the antibiotic susceptibility patterns of *S. pneumoniae* isolates according to the different age groups is also presented in Table 2; no notable differences were observed in terms of resistance between the two age groups.

Furthermore, in total, 15.5% (24/155) bacterial isolates were multi-drug resistant (MDR) (resistant to PEN, SXT, and TET) and 87.5% (21/24) of these MDR isolates were *S. aureus*. In addition, of the 62.5% (15/24) MDR isolates, 58.3% (14/24) were *S. aureus* recovered in infants.

## 3. Discussion

This study aimed to determine the antibiotic susceptibility of *Staphylococcus aureus* and *Streptococcus pneumoniae* isolates obtained from the nasopharynx of febrile children under 5 years in Nanoro, a rural area of Burkina Faso. The study provides evidence for concerning high resistance rates to antibiotics that are commonly used to treat suspected respiratory tract infections and septicemia in Burkina Faso. A significant part of these (sometimes serious) infections are often caused by *S. aureus* and *S. pneumoniae* that are usually considered to be normal colonizers of the nasopharynx of these children [1,2,3]. 

The high resistance rates of *S. aureus* to several commonly used first-line antibiotics are of concern. Particularly, the multidrug resistant S. *aureus* isolates make many antibiotics clinically inefficient and, thereby, reduces treatment options [8,9]. Moreover, a large proportion of methicillin-resistant *S. aureus* (MRSA) was also highly resistant to PEN and TET, and this resistance might be due to the β-lactamases produced by *S. aureus* [5]. In addition, almost all methicillin-sensitive *S. aureus* (MSSA) were resistant to PEN and TET. These findings are of great concern; this was also mentioned by other studies from Burkina Faso [6] and other African countries [18,19,20,21,24]. In contrast, some antibiotics, including gentamicin (GEN) and vancomycin (VAN), are still effective in our study and could serve as alternative treatment options. A limitation of the current study is that the resistance genes of MRSA have not been characterized and we have not established resistance genes similarities between MRSA and MSSA isolates. The study was restricted to phenotypical resistance assessment of the isolated bacteria. More advanced phenotypic (e.g., automated systems) or genotypic (e.g., polymerase chain reaction) methods to determine antibiotic susceptibility are unfortunately still out of reach for many laboratories in low- and middle-income countries (LMICs), including Burkina Faso.

Relatively few *S. pneumoniae* isolates were retrieved in the present study. This low colonization rate could be due to the introduction of the 13-valent pneumococcal conjugate vaccine (PCV-13) into the vaccination program of Burkina Faso in 2013 [12,13]. The high efficacy of this vaccine to reduce *S. pneumoniae* carriage in general was also demonstrated by Kiemde et al. [25], who obtained only three [3] isolates of this species from blood from the same study population. However, high resistance rates to SXT and TET and moderate resistance rate to PEN, which are part of the first-line antibiotics used to treat septicemia and non-severe and severe pneumonia caused by *S. pneumoniae* [26], were observed. This observation is also in line with other studies from Burkina Faso [7,27] and other African countries [28,29,30]. It should be noted that the majority of colonizing *S. pneumoniae* isolates were from infants and part of these children are in the process of receiving the full pneumococcal vaccination course [31]. The colonizing of the nasopharynx by resistant *S. pneumoniae* strains is decreased in older toddlers who have received the full pneumococcal vaccination, confirming the impact of the pneumococcal vaccine after its introduction in the expanded immunization program in Burkina Faso and worldwide [31,32]. 

Although the introduction of the PCV-13 has decreased pneumococcal meningitis incidence in children under 5 in Burkina Faso, it is important to note that some serotypes, such as serotype 1, 23F, 6A/6B, and 12F/12A/12B/44/46, remain predominant in these children [12,13] including other *S. pneumoniae* serotypes that the PCV-13 vaccine does not cover. In particular, serotypes 6A and 23F have been reported to develop multi-drug resistance before their inclusion in the PCV-13 [7], and continue to do so even after their inclusion in the vaccine as suggested by some studies [33,34]. Vaccination might facilitate the introduction of new more resistant pneumococcal serotypes that replace the vaccine serotypes [35,36]. Most likely, these serotypes or those that are not covered by PCV13 are responsible for resistance observed in our study, but this cannot be confirmed as serotyping was not performed.

It is relevant to note that all *S. pneumoniae* strains isolated in this study, albeit, few, were susceptible to ceftriaxone (CRO) and ampicillin (AMP), which are the first-line antibiotics to treat meningitis and suspected septicemia, respectively, in Burkina Faso. Furthermore, the *S. aureus* isolates were susceptible to vancomycin, which is used to treat infections caused by MRSA [37,38]. In addition, our study outcomes support the use of ampicillin for the treatment of suspected pneumonia. Whenever possible, as a mean to curtail resistance, the use of the narrowest spectrum antibiotics is preferred, which is, in this case, AMP.

The present study showed that a significant proportion of *S. aureus* and *S. pneumoniae* isolates from the nasopharynx of young febrile children is resistant to many commonly used antibiotics. This poses a serious problem in the management and treatment of infectious disease. It is, therefore of utmost importance that the spread of (emerging) resistant bacteria in Burkina Faso (and probably the whole West African region) is slowed down and that surveillance structures of antibiotic resistance must be reinforced. In order to manage this resistance problem, more extensive studies on phenotypical and molecular resistance of colonizing bacterial strains from the nasopharynx should be conducted. Knowing that the nasopharynx is a niche propitious to spread resistant strains in communities [2,3,4,21], such studies will provide significant data on the antibiotic resistance and help to refine treatment guidelines at national and global level. 

## 4. Materials and Methods

### 4.1. Study Design and Participants

The present observational study was embedded in a large research project implemented in the Health District of Nanoro (central-west Burkina Faso) and performed from 2014 to 2018 that investigated the etiology, diagnosis, and treatment of fever episodes in children under 5 years [25]. Written informed consent was obtained from the parent or legal guardian prior to enrolment of a child in the study. The study protocol was approved by the National Ethical Committee in Health Research, Burkina Faso (Deliberation N° 2014-11-130). 

Febrile children under the age of 5 years with an axillary temperature ≥37.5 °C presenting at one of the study health facilities were diagnosed according to the International Classification of Diseases in Childhood [39]. They were treated according to the national guidelines for the management of childhood diseases based on the world health organization (WHO) guidelines for the Integrated Management of Childhood Illness [40]. The children were not further follow-up to determine treatment outcome in the context of this study. From each recruited child, different samples were collected, regardless of the potential cause of fever. The clinical specimens were transported to the Microbiology Laboratory of the Clinical Research Unit of Nanoro (CRUN) for microbiology analysis according to the standard operating procedures (SOPs). A standard case record form (CRF) was used to collect details of clinical examinations, diagnosis, and antibiotics prescription. 

### 4.2. Laboratory Procedures

Nasopharyngeal samples were collected with sterile cotton swabs from each participant by trained study nurses. After collection, the nasopharyngeal swabs were inoculated in skim milk, tryptone, glucose, and glycerin broth, and transported to the laboratory where they were processed immediately. Each sample was vortexed and 200 µl was subsequently transferred to 10 mL of Todd–Hewitt broth for enrichment and incubated at 35 ± 2 °C for 24 h. Next, each broth was sub-cultured onto sheep blood agar, and mannitol salt agar (MSA). The sheep blood agar plates were incubated under 5% CO_2_, and the MSA plates under aerobic conditions, at 35 ± 2 °C for 24 h. 

Bacterial isolates were identified using standard microbiology methods [41,42,43]. *S. pneumoniae* were identified by their ability to produce alpha hemolysis on sheep blood agar, and their inability to produce catalase [42]. *S. aureus* was identified as small to large yellowish colonies (ability to ferment mannitol) on MSA plates, and by positive reaction to the catalase and coagulase tests [42]. 

### 4.3. Antimicrobial Susceptibility Testing (AST)

Antimicrobial susceptibility was tested by disk diffusion (Kirby–Bauer) and Epsilometer (*E*-test) methods, and interpreted according to Clinical and Laboratory Standards Institute (CLSI) guidelines [44]. Antibiotics used for susceptibility testing are listed in Table 3. Mueller–Hinton agar with 5% of sheep blood was used for AST of *S. pneumoniae* while Mueller–Hinton agar was used for *S. aureus*. The agar plates were inoculated aseptically with bacterial suspension at McFarland 0.5 (measured by BD PhoenixSpec, Nephelometer Becton Dickinson and Company, Sparks, Maryland, USA) and incubated under either atmospheric condition or 5% of CO_2_ for 18–24 h for *S. aureus* and *S. pneumoniae*, respectively. According to CLSI guidelines, the minimal inhibitory concentration (MIC) as determined by Epsilometer (*E*-test) was used for some antibiotics (see Table 3) [44]. Furthermore, methicillin-resistant *Staphylococcus aureus* (MRSA) strains were phenotypically identified when the diameter of the cefoxitin disc (30 µg) was ≤21 mm [44]. Inducible resistance for both *S. aureus* and *S. pneumoniae* to Clindamycin (CC) was determined by D-testing. 

### 4.4. Quality Control

Standard bacteriological procedures were followed in accordance with the local microbiology standard operating procedures (SOPs) to ensure the reliability of the laboratory results. In addition, all the laboratory processes (culture media, reagents, AST disks and equipment) were quality controlled using American Type Culture Collection (ATCC^®^) standard reference strains. Furthermore, CRUN microbiology laboratory is enrolled to the external quality assessment program of the National Institute for Communicable Diseases (NICD, Johannesburg, South Africa), supported by the World Health Organization (WHO) Africa. 

### 4.5. Data Analysis

Data were entered into Microsoft Excel version 2016, checked by two independent technicians, and subsequently validated by the laboratory manager. The data were then analyzed using STATA^®^ statistical software version 13 StataCorp LLC, College Station, TX, USA. Categorical variables were summarized as proportions and Pearson’s chi-square test were performed. The median was used for continuous variables. A *p* value of <0.05 was considered significant.

Children were stratified in age groups following the expanded national vaccination program of Burkina Faso Ministry of Health (MoH) [31]. The following age categories were made: Neonates: <1 month of age, have not received the pneumococcal vaccine;Infants: ≥1–<30 months of age, in progress of receiving the full course of pneumococcal vaccination;Older toddlers: ≥30–<60 months of age, completed the full course of pneumococcal vaccination (expected to be fully immunized).

A bacterial isolate was considered MDR when it was resistant to at least one antibiotic agent in three antibiotic categories [45].

## 5. Conclusions

This study revealed high resistance rates to antibiotics that are commonly used to treat suspected bacterial respiratory tract infections. Children who received the 13-valent pneumococcal conjugate vaccine carried the highest proportion of resistant bacteria. The research highlights the necessity to perform frequent and extensive antibiotic-resistance studies, including molecular assessment, to ensure that the shrinking arsenal of effective antimicrobials remains effective.

## Ethics Approval and Consent to Participate

A child was only enrolled in the study after obtaining signed written informed consent of her/his parent or legal guardian. The study protocol was reviewed and approved by the National Ethical Committee in Health Research, Burkina Faso (Deliberation No. 2014-11-130).

## Figures and Tables

**Table 1 antibiotics-10-00444-t001:** Study population characteristics.

Characteristic of Children	Study Population	Nasopharyngeal Bacterial Growth				
		Confirmed Bacterial Colonization		Bacterial Species		
				*S. aureus*	*S. pneumoniae*	*S. aureus* + *S. pneumoniae*
Total, *n* (%)	629 (100)	154 (24.5)	*p*-value	148 (96.1)	5 (3.2)	1 (0.6)
Gender			0.55			
Male, *n* (%)	340 (54.1)	80 (23.5)		76 (95.0)	3 (3.8)	1 (1.3)
Female, *n* (%)	289 (45.9)	74 (25.6)		72 (97.3)	2 (2.7)	0 (0)
Age in months, median (IQR)	19 (11.0–32.0)	18 (10–29)		18 (10–29)	11 (9–19)	Not applicable
EPI * status, Yes, *n* (%)	413 (65.7)	100 (24.2)	0.08	97 (97.0)	3 (3.0)	0 (0)
≥1–<30 (M), (Infants), (N = 452), *n* (%)	316 (69.9)	83 (26.3)		81 (97.6)	2 (2.4)	0 (0)
≥30–<60 (M), (Older toddlers), (N = 176), *n* (%)	97 (55.1)	17 (17.5)		16 (94.1)	1 (5.9)	0 (0)

M: month; N: total number; *n*: sub-total number; %: percentage; IQR: interquartile range; EPI: expanded program of immunization; *: The single neonate (age < 1 month was not vaccinated. Nasopharyngeal carriage.

**Table 2 antibiotics-10-00444-t002:** The distribution of antibiotic resistance of *S. aureus* and *S. pneumoniae* isolated from the children’s nasopharynx according to the different age groups.

Type of AB	PEN ^a^	AMP ^a^	GEN	SXT	ERY	OX ^b^	CRO	CIP	TET	CL	CC	VAN	IPM
*S. aureus*													
≥1–<30, (M), (Infants) (N = 112), *n* (%)	107 (95.5)	-	0 (0)	13 (11.6)	19 (17.0)	4 (3.6)	-	5 (4.5)	98 (87.5)	1 (0.9)	14 (12.5)	0 (0)	-
≥ 30–<60, (M), (Older toddlers) (37), *n* (%)	36 (97.3)	-	0 (0)	9 (24.3)	2 (5.4)	2 (5.4)	-	2 (5.4)	33 (89.2)	1 (2.7)	1 (2.7)	0 (0)	-
Total (N = 149)	143 (96.0)	-	0 (0)	22 (14.8)	21 (14.1)	6 (4.0)	-	7 (4.7)	131 (87.9)	2 (1.3)	15 (10.1)	0 (0)	-
*S. pneumoniae*													
≥ 1–<30, (M), (Infants) (N = 4), *n* (%)	2 (50)	0 (0)	-	3 (75.0)	0 (0)	-	0 (0)	-	4 (100)	0 (0)	0 (0)	0 (0)	0 (0)
≥ 30–<60, (M), (Older toddlers) (2), *n* (%)	1 (50)	0 (0)	-	2 (100)	0 (0)	-	0 (0)	-	2 (100)	1 (50)	0 (0)	0 (0)	0 (0)
Total (N = 6)	3 (50)	0 (0)	-	5 (83.3)	0 (0)	-	0 (0)	-	6 (100)	1 (16.7)	0 (0)	0 (0)	0 (0)

AB: antibiotic; M: month; N: total number; *n*: sub-total number; %: percentage; PEN: penicillin; AMP: ampicillin; ^a^: Based on the breakpoints of non-meningitis for S. pneumoniae GEN: gentamicin; SXT: trimethoprim-sulfamethoxazole; ERY: erythromycin; OX: oxacillin; ^b^: The results of oxacillin are reported in this table according to the results of cefoxitin tested as a proxy; CRO: ceftriaxone; CIP: ciprofloxacin; TET: tetracycline; CL: chloramphenicol; CC: clindamycin; VAN: vancomycin; IPM: imipenem.

**Table 3 antibiotics-10-00444-t003:** Antibiotic categories and antibiotic agents used for susceptibility testing.

Antibiotic Categories	Antibiotic Agents	Disc Content	*E*-Test Content
Penicillins	Penicillin (PEN)	-	0.016–256 µg/L
	Cefoxitin (FOX) *	30 µg	-
	Ampicillin (AMP)	-	0.016–256 µg/L
Extended-spectrum cephalosporin; 3rd generation cephalosporin	Ceftriaxone (CRO)	-	0.016–256 mg/L
Fluoroquinolones	Ciprofloxacin (CIP)	5 µg	-
Folate pathway inhibitor	Trimethoprim-sulfamethoxazole (SXT)	1.25/23.75 µg	-
Aminoglycosides	Gentamicin (GEN)	10 µg	-
Macrolides	Azithromycin (AZI)	15 µg	-
	Erythromycin (ERY)	15 µg	
Phenicols	Chloramphenicol (CL)	30 µg	-
Carbapenems	Imipenem (IPM)	-	0.02–32 mg/L
Lincosamides	Clindamycin (CC)	2 µg	
Glycopeptides	Vancomycin (VAN)	30 µg	0.016–256 µg/L
Tetracyclines	Tetracycline (TET)	30 µg	

*: Cefoxitin disc test used as a proxy test for oxacillin; *E*-test: Epsilometer

## Data Availability

Data is contained within the article. Datasets used and/or analyzed in this study are available from the corresponding author upon reasonable request. WHO Africa is thanked for supporting the external quality assessment of the laboratory. Standard reference strains *Staphylococcus aureus* ATCC 25923, *Streptococcus pneumoniae* ATCC 49619, *Staphylococcus epidermidis* ATCC 14990, *Escherichia coli* ATCC-25922, *Streptococcus pyogenes* ATCC 19615, *Enterococcus faecalis* ATCC 29212 were obtained from The American Type Culture Collection.

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
