# Peer review of "Antibiotic Susceptibility of Staphylococcus aureus and Streptococcus pneumoniae Isolates from the Nasopharynx of Febrile Children under 5 Years in Nanoro, Burkina Faso"

_antibiotics, 2021, doi:10.3390/antibiotics10040444_

Round 1
Reviewer 1 Report
In this manuscript, the authors investigated drug resistance S. aureus and S. pneumonia after introducing a vaccine against S. pneumonia in Burkina Faso. The results suggested that MRSA has still been spread in the infant 0-5 year’s children and rarely colonized S. pneumonia in the nasopharynx. This research is valuable for AMR researchers, and continuous effort is required to reduce AMR. I want to suggest the authors following minor correction.
- The authors described that the reduction of isolation of S. pneumonia strain compared before vaccination. It is better to cite and compare actual figures with previous isolation rate with similar research on children using actual numbers.
Author Response
Response to reviewer comments
Reviewer 1 comments:
Point 1: The authors described that the reduction of isolation of S. pneumonia strain compared before vaccination. It is better to cite and compare actual figures with previous isolation rate with similar research on children using actual numbers.
Response 1: We are not completely sure what the reviewer suggests here. In the introduction (lines 81-84) we have presented the actual numbers of reduction found in Burkina Faso. Our study further shows the efficacy of the PCV-13 vaccine as relatively few S. pneumonia could be isolated. This is in line with a recent observation in blood in the same study area. We have now also mentioned this in the discussion section (see lines: 225-228).
Reviewer 2 Report
The manuscript deals with an essential topic of species, the carrier of which may cause symptoms that are dangerous to life. The subject is already quite extensively described in the literature, but taking into account this country is essential from the local medical care perspective.
For example, the methods were thoughtful and appropriate by multiplying the bacteria in the broth just before being inoculated onto solid media, however, a few things should be improved. The above work still requires much work. First of all, no statistical analysis was performed, which from the point of view of such a large population, even taking into account gender (male-female; 54% -46%), is very important and certainly useful. Moreover, the authors themselves emphasize (lines 182-183) that the strains may produce beta-lactamase. I believe that taking into account the availability of methods, and it should be verified, for example, genetically, using PCR technique. The authors in the text (lines 305-310) describe the study groups' selection and inclusion criteria; three study groups are described. However, it is not consistent with the information presented in the tables above in the text.
Most importantly, there is no control group, such as children with no fever, which would allow us to draw reliable conclusions from the research. It was also not indicated on what days and how long the children had a fever. There are also mistakes e.g. in lines 106, 229, and many double spaces or lack of italics in the case of microorganisms (this is not standardized in the text).
Most of the literature that the authors refer to is more than five years old, and for this journal, this is of importance.
Author Response
Response to reviewer comments
Reviewer 2 comments:
Point 1: First of all, no statistical analysis was performed, which from the point of view of such a large population, even taking into account gender (male-female;54% -46%), is very important and certainly useful.
Response 1: We have now applied Chi-squared statistics for differences to determine the significance of observed differences between groups (gender and age group) (see table 2, line 144-145 and lines 362-365). This is to our opinion the only statistics that can be meaningfully done on the data we have presented. The data on resistance (expressed as a percentage) are isolated data and do not have associations which each other. We have now included this analysis in the manuscript where applicable.
Point 2: Moreover, the authors themselves emphasize (lines 182-183) that the strains may produce beta-lactamase. I believe that taking into account the availability of methods, and it should be verified, for example, genetically, using PCR technique.
Response 2: We agree with the reviewer that genetic methods could be applied. However, we would like to emphasize that the work was conducted in Burkina Faso, one of the poorest countries in the world with limited research infrastructure. Access to modern genotyping methods is unfortunately limited. We hope that even with this limitation the reviewer appreciates our efforts to conduct a study on bacteria resistance isolated from febrile children’s nasopharynx in our country. We have now included the following section in the discussion (see lines 216-222) to accommodate the reviewer’s comment: The study was restricted to a phenotypic assessment of the isolated bacteria. More advanced phenotypic (e.g., automated systems) or genotypic (e.g., polymerase chain reaction) methods to determine antibiotic susceptibility are unfortunately still out of reach for many laboratories in LMICs, including Burkina Faso.
Point 3: The authors in the text (lines305-310) describe the study groups' selection and inclusion criteria; three study groups are described. However, it is not consistent with the information presented in the tables above in the text.
Response 3: Indeed, we have defined three age groups:
- Neonates: < 1month of age
- Infants: ≥1 - <30 months of age
- Older toddlers: ≥30 – <60 months of age
And this categorization is used throughout the manuscript, including the tables. It is noted that the study recruited only 1 (one) neonate. Bacterial colonization was not observed in this single neonate and there are no further data presented of this child in this paper, including the tables. We have now indicated this in the text (see lines: 142-143). Thus, the tables are correct by presenting just two categories.
Furthermore, we have made the age categories clearer in the tables by adding the appropriate symbols “≥” or “<” to the different categories in Tables 2 and 3.
Point 4: There is no control group, such as children without fever.
Response 4: The study is not a cross-sectional survey, but an observational study. We have only recruited children who attended the study health facilities (practical reason – as we did the study in a remote rural area) and these children did have a fever. We did not seek a control group in the context of the present study, neither did we do a follow-up of cases. This information is now inserted in the manuscript (see lines: 289-290 and 293-295) As we are mostly interested in resistance monitoring, we think that this approach is appropriate.
Point 5: It was also not indicated on what days and how long the children had a fever.
Response 5: All children had an axillary temperature > 37.5° C, at enrolment (this was an inclusion criterion; we have now indicated this in the manuscript). There was in the context of the current study no follow-up of children. This information is now included in the manuscript (see lines: 289-290 and 293-295).
Point 6: There are also mistakes e.g., in lines 106, 229
Response 6: Thank you for this observation and we have corrected this.
Point 7: Many double spaces.
Response 7: We have re-checked the manuscript and removed unneeded double spaces.
Point 8: Lack of italics in the case of microorganisms (this is not standardized in the text).
Response 8: We have re-checked the text and modified it accordingly.
Point 9: Most of the literature that the authors refer to is more than five years old,
Response 9: The manuscript contains 46 references of whom none are older than 2000. So,
- 14 (30.4%) papers are from the period 2000-2010;
- 17 (37.0%) from the period 2011-2015; and
- 15 (32.6%) from 2015- 2020.
We think that is a nice representation and ensures that the information contained within the manuscript is up to date.
Round 2
Reviewer 2 Report
The authors corrected the manuscript, all suggestions were taken into account during manuscript improvement.